# Trajectory Attention for Fine-grained Video Motion Control

**Zeqi Xiao**[1], **Wenqi Ouyang**[1], **Yifan Zhou**[1],
**Shuai Yang**[2], **Lei Yang**[3], **Jianlou Si**[3], **Xingang Pan**[1]
[1]S-Lab, Nanyang Technological University,
[2]Wangxuan Institute of Computer Technology, Peking University
[3]SenseTime Research
{zeqi001, yifan006, wenqi.ouyang, xingang.pan}@ntu.edu.sg
williamyang@pku.edu.cn
{jianlousi,leiyang}@sensetime.com

## Abstract

Recent advancements in video generation have been greatly driven by video diffusion models, with camera motion control emerging as a crucial challenge in creating view-customized visual content. This paper introduces trajectory attention, a novel approach that performs attention along available pixel trajectories for fine-grained camera motion control. Unlike existing methods that often yield imprecise outputs or neglect temporal correlations, our approach possesses a stronger inductive bias that seamlessly injects trajectory information into the video generation process. Importantly, our approach models trajectory attention as an auxiliary branch alongside traditional temporal attention. This design enables the original temporal attention and the trajectory attention to work in synergy, ensuring both precise motion control and new content generation capability, which is critical when the trajectory is only partially available. Experiments on camera motion control for images and videos demonstrate significant improvements in precision and long-range consistency while maintaining high-quality generation. Furthermore, we show that our approach can be extended to other video motion control tasks, such as first-frame-guided video editing, where it excels in maintaining content consistency over large spatial and temporal ranges.

## 1 Introduction

Video generation has experienced remarkable advancements in recent years, driven by sophisticated deep learning models such as video diffusion models and temporal attention mechanisms (OpenAI, 2024; Chen et al., 2024; Wang et al., 2023a; Guo et al., 2023b). These innovations have enabled the synthesis of increasingly realistic videos, fueling fields in areas such as filmmaking (Zhao et al., 2023; Zhuang et al., 2024) and world modeling (OpenAI, 2024; Valevski et al., 2024). Video motion control, which aims to produce customized motion in video generation, has emerged as a crucial aspect (Yang et al., 2023b; Ling et al., 2024; Ouyang et al., 2024; Ku et al., 2024; Zhao et al., 2023).

Among various control signals, camera motion control has garnered increasing attention due to its wide applications in creating view-customized visual content. However, effectively conditioning generation results on given camera trajectories remains non-trivial. Researchers have explored several approaches to address this challenge. One method involves encoding camera parameters into embeddings and injecting them into the model via cross-attention or addition (Wang et al., 2024c; He et al., 2024; Bahmani et al., 2024). While straightforward, this approach often yields imprecise and ambiguous outputs due to the high-level constraints and implicit control mechanisms it employs. Another strategy involves rendering partial frames based on camera trajectories and using these either as direct input (Hu et al., 2024; Yu et al., 2024) or as optimization targets (You et al., 2024) for frame-wise conditioning. Although this method provides more explicit control, it often neglects temporal correlations across frames, leading to inconsistencies in the generated sequence.

---

Project page at this URL.

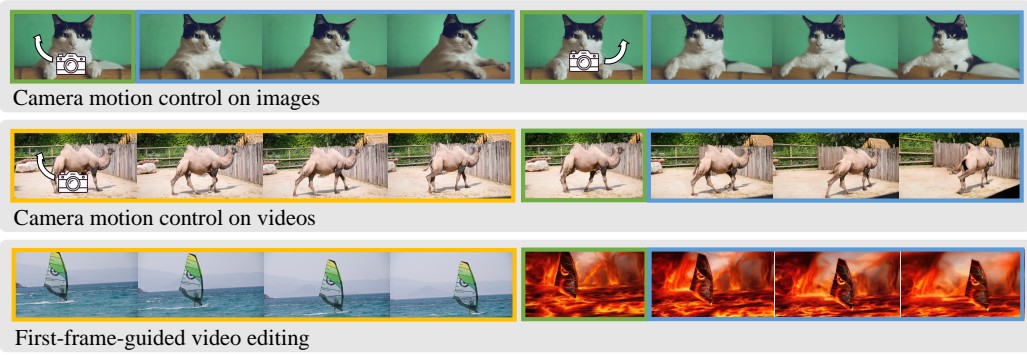

Figure 1: **Trajectory attention** injects partial motion information by making content along trajectories consistent. It facilitates various tasks such as camera motion control on images and videos, and first-frame-guided video editing. Yellow boxes indicate reference contents. Green boxes indicate input frames. Blue boxes indicate output frames.

In response to these limitations, recent methods have begun to address temporal relationships by leveraging 3D inductive biases (Xu et al., 2024; Li et al., 2024). These approaches focus on narrowed domains, utilizing specific settings such as row-wise attention (Li et al., 2024) or epipolar constraint attention (Xu et al., 2024). As we consider the trajectory of a camera moving around scenes, it becomes apparent that certain parts of the moving trajectories of pixels, represented as a sequence of 2D coordinates across frames, are predictable due to 3D consistency constraints. This observation raises an intriguing question: can we exploit such trajectories as a strong inductive bias to achieve more fine-grained motion control?

Revisiting the temporal attention mechanism, which is central to video models for synthesizing dynamic motions with consistent content, we can view the dynamics as pixel trajectories across frames. The temporal attention mechanism, with its generic attention design, functions by *implicitly* synthesizing and attending to these trajectories. Building on this observation, when parts of the trajectories are available, the attention along these trajectories can be modeled *explicitly* as a strong inductive bias to produce controlled motion with consistent content.

To this end, we propose trajectory attention that performs attention along the available trajectories across frames for fine-grained camera motion control. Instead of directly adapting the temporal attention to operate on trajectories, which yields suboptimal results in practice, we model trajectory attention as an auxiliary branch alongside the original temporal attention. This design is critical due to the distinct goals of these two attention mechanisms. Temporal attention, which must balance motion synthesis and content consistency, typically focuses on short-range dynamics and attends to adjacent frames within a local window. In contrast, trajectory attention is designed to ensure long-range consistency across features along a trajectory (see Fig. 2). The trajectory attention branch can inherit the parameters of the original temporal attention for efficient tuning, and its output is added to the output of temporal attention as residuals. This whole design offers several merits: 1) it allows better division of tasks: trajectory attention manages motion control and ensures long-range consistency along specified paths, while temporal attention synthesizes motion for the rest regions; 2) it can integrate seamlessly without modifying the original parameters; 3) it supports sparse trajectories, as the condition is injected moderately, meaning available trajectories do not have to cover all pixels.

Our experiments on camera motion control for images and videos demonstrate that our designs significantly enhance precision and long-range consistency. As shown in Fig. 1, our approach leverages a stronger inductive bias that optimizes the attention mechanism. This results in improved control precision while maintaining high-quality generation. The proposed trajectory attention can be extended to other video motion control tasks, such as first-frame-guided video editing. Existing techniques often struggle to maintain content consistency over large spatial and temporal ranges (Ku et al., 2024; Ouyang et al., 2024). In contrast, our method's ability to model long-range, consistent correspondences achieves promising results in these challenging scenarios. Moreover, the efficiency of our design allows for training with limited data and computational resources, making it generalizable to diverse contexts and frame ranges.

## 2 RELATED WORKS

**Video Diffusion Models.** The field of video generation has seen significant advancements in recent years, especially in the area of video diffusion models (Ho et al., 2022; Guo et al., 2023b; Chen et al., 2023a; Wang et al., 2023b;a; OpenAI, 2024; Blattmann et al., 2023; Guo et al., 2023a; Chen et al., 2024; Hong et al., 2022).

The core of motion modeling of video diffusion models is the temporal attention module. Some approaches (Guo et al., 2023b; Chen et al., 2023a; Wang et al., 2023b;a) decompose attention into spatial and temporal components, where temporal attention aligns features across different frames. Others (Yang et al., 2024b; OpenAI, 2024; Lab & etc., 2024) integrate spatial and temporal attention into a unified mechanism, capturing both types of information simultaneously. While these methods rely on data-driven techniques to implicitly learn dynamic video priors within the attention mechanism, how to leverage such priors for explicit and precise motion control remains under-explored.

**Motion Control in Video Generation.** Prior works have explored various control signals for video motion control (Guo et al., 2024; Niu et al., 2024; Yu et al., 2023; Chen et al., 2023b; Yang et al., 2024a; Zuo et al., 2024; Zhu et al., 2024a; Zhao et al., 2023; Chen et al., 2023c; Zhang et al., 2023b), including sketches (Wang et al., 2024b), depth maps (Wang et al., 2024b), drag vectors (Yin et al., 2023; Teng et al., 2023; Deng et al., 2023), human pose (Zhang et al., 2024; Zhu et al., 2024b), object trajectory (Qiu et al., 2024; Wang et al., 2024a; Wu et al., 2024; Gu et al., 2024), and features extracted from reference videos (Yatim et al., 2023; Xiao et al., 2024; Yang et al., 2023b; Ouyang et al., 2024; Ku et al., 2024).

One important branch of video motion control is camera motion control, also known as novel view synthesis. In this regard, Wang et al. (2024c); He et al. (2024); Bahmani et al. (2024); Wu et al. (2024) utilize high-level condition signals by encoding camera pose parameters into conditional features. However, these methods often lack precision in capturing detailed temporal dynamics, as they impose weak constraints on the resulting motion. Hou et al. (2024) enables camera control by rendering incomplete warped views followed by re-denoising. Müller et al. (2024); Yu et al. (2024); You et al. (2024) render partial videos as guidance and leverage video generation models to inpaint the remaining frames. Despite these innovations, their approaches suffer from temporal inconsistency due to the lack of consideration for sequential coherence. Methods such as those proposed by Shi et al. (2024); Xu et al. (2024); Cong et al. (2023); Kuang et al. (2024) explicitly modify attention using optical flow or epipolar constraints. These solutions can be viewed as a weaker variant of trajectory-consistent constraint. Our approach introduces a trajectory attention mechanism for motion information injection. Such a mechanism imposes a strong inductive bias on the temporal dimension, as also explored by Patrick et al. (2021) for video recognition. By leveraging the attention mechanism, our method affords precise control over video generation and improves efficiency—all without requiring specially annotated datasets, such as camera pose annotations. This approach enhances motion control throughout the generation process while preserving the fidelity of temporal dynamics.

## 3 METHODOLOGY

This section introduces trajectory attention for fine-grained motion control. We first outline video diffusion models with a focus on temporal attention (Sec. 3.1), then adapt it for trajectory attention and discuss its limitations (Sec. 3.2). We present trajectory attention as an additional branch, with visualizations of its effectiveness (Sec. 3.3), and describe an efficient training pipeline (Sec. 3.4).

### 3.1 PRELIMINARY

The core of video motion modeling lies in the temporal attention mechanism within video diffusion models, whether applied through decomposed spatial and temporal attention or full 3D attention, to capture robust motion priors. This paper demonstrates the decomposed setting, which is more widely used and has greater open-source availability. However, our design is also adaptable to full 3D attention, as will shown in the experimental results and appendix.

A typical video diffusion architecture for decomposed spatial-temporal attention includes convolutional layers, spatial attention blocks, and temporal attention blocks. The temporal attention operates

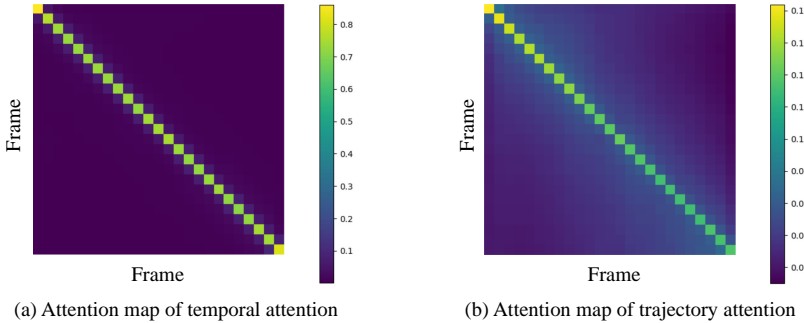

(a) Attention map of temporal attention       (b) Attention map of trajectory attention

Figure 2: **Attention map visualization of temporal attention and trajectory attention.** (a) Temporal attention tends to concentrate its weight on a narrow, adjacent frame window. (b) In contrast, trajectory attention exhibits a broader attention window, highlighting its capacity to produce more consistent and controllable results. Here, the attention map is structured with the frame number as the side length. The attention weights are normalized within the range of 0 to 1, where higher values (indicated by light yellow) represent stronger attention.

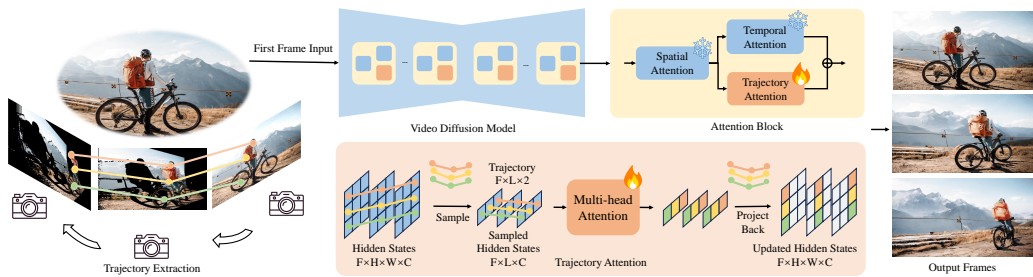

Figure 3: **Overview of the proposed motion control pipeline.** Our method allows for conditioning on trajectories from various sources – such as camera motion derived from a single image, as shown in this figure. We inject these conditions into the model through trajectory attention, enabling explicit and fine-grained control over the motion in the generated video.

as follows. Given an input latent feature $\mathbf{Z} \in \mathbb{R}^{F \times H \times W \times C}$, where $F$, $H$, $W$, and $C$ represent the number of frames, height, width, and channels, respectively, temporal attention operates along the frame dimension. The feature $\mathbf{Z}$ is first projected into query ($\mathbf{Q}$), key ($\mathbf{K}$), and value ($\mathbf{V}$):

$$\mathbf{Q} = p_q(\mathbf{Z}), \mathbf{K} = p_k(\mathbf{Z}), \mathbf{V} = p_v(\mathbf{Z}), \tag{1}$$

where $p_q$, $p_k$, and $p_v$ are learnable projection functions. Temporal attention is then applied along the frame dimension as:

$$\mathbf{Z}' = \text{Softmax}(\mathbf{Q}\mathbf{K}^T)\mathbf{V}, \tag{2}$$

yielding the output latent feature $\mathbf{Z}'$. For simplicity, we omit the details like rescaling factor and multi-head operations. With large-scale training, temporal attention effectively captures dynamic and consistent video motions, making it a natural candidate for motion control in video models.

## 3.2 TAMING TEMPORAL ATTENTION FOR TRAJECTORY ATTENTION

As shown in Fig. 4, vanilla temporal attention operates on the same spatial position across different frames, where the coordinates in the attention form predefined trajectories across frames.

Since temporal attention has already learned to model motion along pre-defined trajectories, a natural extension is to tame temporal attention for additional trajectory attention. For example, given a set of trajectories $\mathbf{Tr}$, where each trajectory is represented by a series of coordinates, we incorporate them into the temporal attention mechanism.

---

**Algorithm 1:** Trajectory-based sampling

---

**Input:** Hidden states $\mathbf{Z} \in \mathbb{R}^{F \times H \times W \times C}$, where $F$ is the number of frames, $H, W$ are the spatial dimensions, and $C$ is the number of channels. $L$ trajectories $\mathbf{Tr} \in \mathbb{R}^{L \times F \times 2}$, where each trajectory specifies $F$ 2D locations. Trajectory masks $\mathbf{M} \in \mathbb{R}^{F \times L}$, where $M_{f,l} \in \{0, 1\}$ indicates whether a trajectory is valid at frame $f$ for trajectory $l$.

1 **foreach** trajectory $i = 1, \ldots, L$ **do**
2      Sample hidden states $\mathbf{Z}_i = \{\mathbf{Z}_f(x_{f,i}, y_{f,i}) \mid f = 1, \ldots, F\} \in \mathbb{R}^{F \times C}$
3      where $(x_{f,i}, y_{f,i})$ are the 2D coordinates from $\mathbf{Tr}[i]$ for each frame $f$.
4 **end**
5 Stack sampled hidden states: $\mathbf{Z}_s = \text{Stack}(\mathbf{Z}_i \mid i = 1, \ldots, L) \in \mathbb{R}^{F \times L \times C}$
6 Mask out invalid hidden states using $\mathbf{M}$: $\mathbf{Z}_t = \mathbf{Z}_s \odot \mathbf{M}$

**Output:** Masked sampled hidden states $\mathbf{Z}_t \in \mathbb{R}^{F \times L \times C}$

---

However, this straighwarpward adaptation often yields suboptimal results due to a conflict between temporal and trajectory attention. Temporal attention is designed to ensure consistency along the trajectory while preserving the dynamism of feature representations. However, achieving both perfectly is challenging. Consequently, temporal attention often prioritizes natural dynamics at the expense of long-range consistency. This is evident in the attention statistics: as shown in Fig. 2(a), the learned temporal attention predominantly focuses on adjacent frames.

In contrast, trajectory attention, given its known dynamics, aims solely to align features along the trajectory. This singular focus on alignment often clashes with the broader objectives of temporal attention. Simply adapting temporal attention to accommodate trajectory information can therefore introduce conflicts. Experimental results further demonstrate that, even with extensive training, the quality of motion control remains suboptimal when trajectory attention is naively integrated.

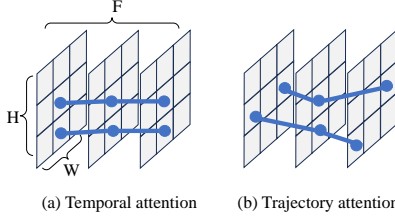

(a) Temporal attention     (b) Trajectory attention

Figure 4: **Visualization of vanilla temporal attention and trajectory attention.**

### 3.3 MODELING TRAJECTORY ATTENTION AS AN AUXILIARY BRANCH

The above analysis reveals that temporal attention and trajectory attention should not share the same set of weights. Inspired by the recent success of Zhang et al. (2023a), we model temporal attention and trajectory attention into a two-branch structure, where trajectory attention is responsible for injecting fine-grained trajectory consistent signal to the origin generation process.

As illustrated in Fig. 3, trajectory attention and temporal attention share the same structure, as well as identical input and output shapes. The key difference lies in the process: we first use the given trajectories to sample features from the hidden states (Algorithm 1), then apply multi-head attention with distinct parameters, and finally project the results back to the hidden state format after frame-wise attention (Algorithm 2).

To validate the purpose distinction, we compare the attention maps (softmax scores along the frame axis) of temporal and trajectory attention, based on the SVD model (Blattmann et al., 2023). As shown in Fig. 2(a) and (b), trajectory attention clearly provides a broader attention window, enabling more consistent and controllable results.

### 3.4 TRAINING TRAJECTORY ATTENTION EFFICIENTLY

As illustrated in Fig. 5, we initialize the weights of the QKV projectors with those from temporal attention layers to harness the motion modeling capabilities learned from large-scale data. Additionally, the output projector is initialized with zero weights to ensure a gradual training process.

The training objective follows the standard approach used in fundamental generation models. For instance, in the case of Stable Video Diffusion (Blattmann et al., 2023), the objective is:

---

**Algorithm 2:** Back projection

---

**Input:** Hidden states after attention $\mathbf{Z}_t' \in \mathbb{R}^{F \times L \times C}$. $L$ trajectories $\mathbf{Tr} \in \mathbb{R}^{L \times F \times 2}$. Trajectory masks $\mathbf{M} \in \mathbb{R}^{F \times L}$.

1 **Initialize**:

$$\mathbf{Z}_p \in \mathbb{R}^{F \times H \times W \times C}, \quad \mathbf{U} \in \mathbb{R}^{F \times H \times W}, \quad \mathbf{Z}_p = \mathbf{0}, \quad \mathbf{U} = \mathbf{0}$$

where $H$ and $W$ are the height and width of the spatial grid.

2 **foreach** $i = 1, \ldots, L$ **do**

3      Add $\mathbf{Z}_t'[i] \in \mathbb{R}^{F \times C}$ to $\mathbf{Z}_p$ at locations $(x_{f,i}, y_{f,i})$ from $\mathbf{Tr}[i]$: $\mathbf{Z}_p(f, x_{f,i}, y_{f,i}, :)\mathrel{+}=\mathbf{Z}_t'[i](f, :)$

4      Update count table $\mathbf{U}$ at the same locations: $\mathbf{U}(f, x_{f,i}, y_{f,i})\mathrel{+}=\mathbf{M}[f, i]$

5 **end**

6 Normalize $\mathbf{Z}_p$ element-wise for valid positions ($\mathbf{U} > 0$):

$$\mathbf{Z}_p(f, x, y, :) = \frac{\mathbf{Z}_p(f, x, y, :)}{\mathbf{U}(f, x, y)} \quad \text{for all} \quad (f, x, y) \quad \text{where} \quad \mathbf{U}(f, x, y) > 0$$

**Output:** Back-projected hidden states $\mathbf{Z}_p \in \mathbb{R}^{F \times H \times W \times C}$

---

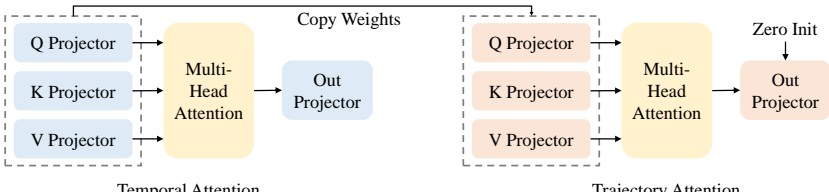

Figure 5: **Training strategy for trajectory attention.** To leverage the motion modeling capability learned from large-scale data, we initialize the weights of the QKV projectors with those from temporal attention layers. Additionally, the output projector is initialized with zero weights to ensure a smooth and gradual training process.

$$\mathbb{E}[||D_\theta(\mathbf{x}_0 + \mathbf{n}; \sigma, \mathbf{c}) - \mathbf{x}_0||_2^2], \tag{3}$$

where $D_\theta$ represents the neural network, $\mathbf{x}_0$ denotes the latent features of the target videos, $\mathbf{n}$ is the noise, $\mathbf{c}$ is the condition signal, and $\sigma$ is the variance parameter.

## 4 FINE-GRAINED CONTROL OF VIDEO GENERATION

This section delves into the process of extracting trajectories for different task settings. While our primary focus is on camera motion control for both static images and dynamic video content, we also showcase the process of trajectory extraction for video editing.

### 4.1 CAMERA MOTION CONTROL ON IMAGES

Algorithm 3 outlines the process of extracting trajectories, denoted as $\mathbf{Tr}$, along with the corresponding validity mask $\mathbf{M}$ from a single image. Unlike prior approaches (Wang et al., 2024c; He et al., 2024), which rely on high-level control signals for video manipulation, our method explicitly models camera motion as trajectories across frames. This enables precise and accurate control of camera movement.

### 4.2 CAMERA MOTION CONTROL ON VIDEOS

The process for camera motion control on videos is more complex than the process for images since the video itself has its own motion. We need to extract the original motion with point trajectory estimation methods like Karaev et al. (2023), then combine the original motion with camera motion to get the final trajectories. We show the details in Algorithm 4.

---

**Algorithm 3:** Trajectory extraction from single image

---

**Input:** Image $\mathbf{I} \in \mathbb{R}^{H_p \times W_p \times 3}$, A set of camera pose with intrinsic and extrinsic parameters,$\{\mathbf{K} \in \mathbb{R}^{3\times3}\}$ and $\{\mathbf{E}[\mathbf{R}; \mathbf{t}]\}$, where $\mathbf{R} \in \mathbb{R}^{3\times3}$ representations the rotation part of the extrinsic parameters, and $\mathbf{t} \in \mathbb{R}^{3\times1}$ is the translation part. The length of the camera pose equals frame number $F$. $H_p$ and $W_p$ are the height and width of the pixel space

1 Estimate the depth map $\mathbf{D} \in \mathbb{R}^{H_p \times W_p}$ from $\mathbf{I}$ given camera pose parameters.
2 Get the translation of pixels $\mathbf{T} \in \mathbb{R}^{F \times H_p \times W_p \times 2}$ based on $\mathbf{I}$ using using $\mathbf{D}$, $\mathbf{K}$, and $\mathbf{E}$.
3 Get trajecories $\mathbf{Tr} = \mathbf{T} + \mathbf{C}$, where $\mathbf{C} \in \mathbb{R}^{H_p \times W_p \times 2}$ is pixel-level grid coordinates of image with shape $H_p \times W_p$.
4 Get valid trajectory mask $\mathbf{M}$ for pixels that within the image space.
**Output:** Trajectories $\mathbf{Tr}$, Trajectory Masks $\mathbf{M}$

---

---

**Algorithm 4:** Trajectory extraction from video

---

**Input:** Video Frames $\mathbf{V} \in \mathbb{R}^{F \times H_p \times W_p \times 3}$, A set of camera pose with intrinsic and extrinsic parameters,$\{\mathbf{K} \in \mathbb{R}^{3\times3}\}$ and $\{\mathbf{E}[\mathbf{R}; \mathbf{t}]\}$. The lenght of camera pose equals to frame number $F$

1 Estimate the depth map $\mathbf{D} \in \mathbb{R}^{F \times H_p \times W_p}$ from $\mathbf{V}$ given camera pose parameters.
2 Estimate point trajecotries $\mathbf{P} \in \mathbb{R}^{F \times L \times 2}$ and the corresponding occlusion masks $\mathbf{M}_o$.
3 Get the translation of pixels $\mathbf{T} \in \mathbb{R}^{F \times H_p \times W_p \times 2}$ using $\mathbf{D}$, $\mathbf{K}$ and $\mathbf{E}$.
4 Sample the translation of point trajectories $\mathbf{P}_t \in \mathbb{R}^{F \times L \times 2}$ from $\mathbf{T}$ using $\mathbf{P}$.
5 Get trajecories $\mathbf{Tr} = \mathbf{P}_t + \mathbf{P}$.
6 Get valid trajectory mask $\mathbf{M} = \mathbf{M}_i \wedge \mathbf{M}_o$, where $\mathbf{M}_i$ is for pixels that within the image space.
**Output:** Trajectories $\mathbf{Tr}$, Trajectory Masks $\mathbf{M}$

---

### 4.3 VIDEO EDITING

Video editing based on an edited first frame has gained popularity recently (Ouyang et al., 2024; Ku et al., 2024). The goal is to generate videos where the content of the first frame aligns with the edited version while inheriting motion from reference videos. Our method is well-suited for this task, as we leverage Image-to-Video generation models that use the edited first frame as a conditioning input while incorporating trajectories extracted from the original videos to guide the motion.

## 5 EXPERIMENTS

### 5.1 EXPERIMENTAL SETTINGS

**Datasets.** We use MiraData (Ju et al., 2024) for training, a large-scale video dataset with long-duration videos and structured captions, featuring realistic and dynamic scenes from games or daily life. We sample short video clips and apply Yang et al. (2023a) to extract optical flow as trajectory guidance. In total, we train with 10k video clips.

**Implementation Details.** We conducted our main experiments using SVD (Blattmann et al., 2023), employing the Adam optimizer with a learning rate of 1e-5 per batch size, with mixed precision training of fp16. We only fine-tune the additional trajectory attention modules which inherit weights from the temporal modules. Our efficient training design allows for approximately 24 GPU hours of training (with a batch size of 1 on a single A100 GPU over the course of one day). We train trajectory attention on the 12-frame video generation modules and apply the learned trajectory attention to both 12-frame and 25-frame video generation models. Despite being trained on 12-frame videos, the trajectory attention performs effectively when integrated into the 25-frame model, demonstrating the strong generalization capability of our design.

**Metrics.** We assessed the conditional generation performance using four distinct metrics: (1) Absolute Trajectory Error (ATE) (Goel et al., 1999), which quantifies the deviation between the estimated and actual trajectories of a camera or robot; and (2) Relative Pose Error (RPE) (Goel et al., 1999),

Table 1: Qualitative comparison on image camera motion control. *: MotionI2V uses AnimateDiff (Guo et al., 2023b) while we use SVD (Blattmann et al., 2023) as the base models. Other methods use SVD as default.

| Setting | Methods | ATE (m, ↓) | RPE trans (m, ↓) | RPE Rot (deg, ↓) | FID (↓) |
|---|---|---|---|---|---|
| 14 frames | MotionCtrl | 1.2151 | 0.5213 | 1.8372 | **101.3** |
| | Ours | **0.0212** | **0.0221** | **0.1151** | 104.2 |
| 16 frames | MotionI2V* | 0.0712 | 0.0471 | 0.2853 | 124.1 |
| | Ours | **0.0413** | **0.0241** | **0.1231** | **108.7** |
| 25 frames | CameraCtrl | 0.0411 | 0.0268 | 0.3480 | 115.8 |
| | NVS_Solver | 0.1216 | 0.0558 | 0.4785 | 108.5 |
| | Ours | **0.0396** | **0.0232** | **0.1939** | **103.5** |

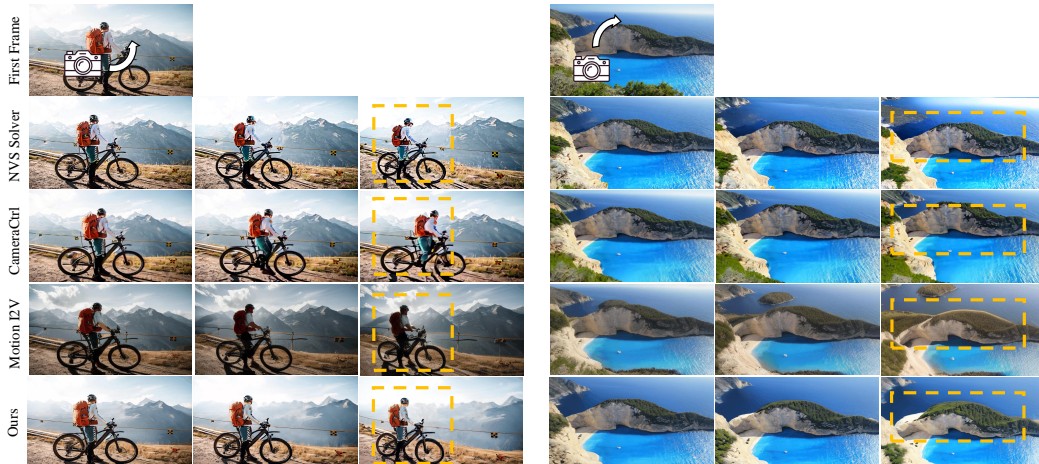

Figure 6: **Qualitative comparisons for camera motion control on images**. While other methods often exhibit significant quality degradation or inconsistencies in camera motion, our approach consistently delivers high-quality results with precise, fine-grained control over camera movements. Regions are highlighted in yellow boxes to reveal camera motion. For a more comprehensive understanding, we highly recommend viewing the accompanying videos in the supplementary materials.

which captures the drift in the estimated pose by separately calculating the translation (RPE-T) and rotation (RPE-R) errors. (3) Fréchet Inception Distance (FID) (Heusel et al., 2017), which evaluates the quality and variability of the generated views.

## 5.2 CAMERA MOTION CONTROL ON SINGLE IMAGES

We compare the results of camera motion control on single images with the methods proposed by Wang et al. (2024c); Shi et al. (2024); He et al. (2024). The evaluation is based on 230 combinations of diverse scenes and camera trajectories. To ensure a fair comparison, our model is tested under varying settings due to the frame limitations of certain models (i.e., (Wang et al., 2024c) only releases a 12-frame version).

Table 1 summarizes the results, showing that our methods consistently achieve higher or comparable control precision in terms of ATE and RPE, along with strong fidelity as measured by FID, compared to other methods (Wang et al., 2024c; Shi et al., 2024; He et al., 2024; You et al., 2024). Although MotionCtrl (Wang et al., 2024c) generates slightly better results in terms of FID, it compromises significantly on control precision. Motion-I2V Shi et al. (2024), which uses flow-based attention, only allows frames to attend to the first frame, leading to quality issues in some cases. In contrast, our approach maintains better control precision while preserving generation quality. It also performs better over longer time ranges than other recent methods (He et al., 2024; You et al., 2024).

We further provide qualitative results in Fig. 6, which is aligned with the conclusions in Table 1.

Table 2: Qualitative comparison on video camera motion control.

| Methods | ATE (m, ↓) | RPE trans (m, ↓) | RPE Rot (deg, ↓) | FID (↓) |
|---|---|---|---|---|
| NVS_Solver | 0.5112 | 0.3442 | 1.3241 | 134.5 |
| Ours | 0.3572 | 0.1981 | 0.7889 | 129.3 |
| Ours (w. NVS_Solver) | **0.3371** | **0.1972** | **0.6241** | **112.2** |

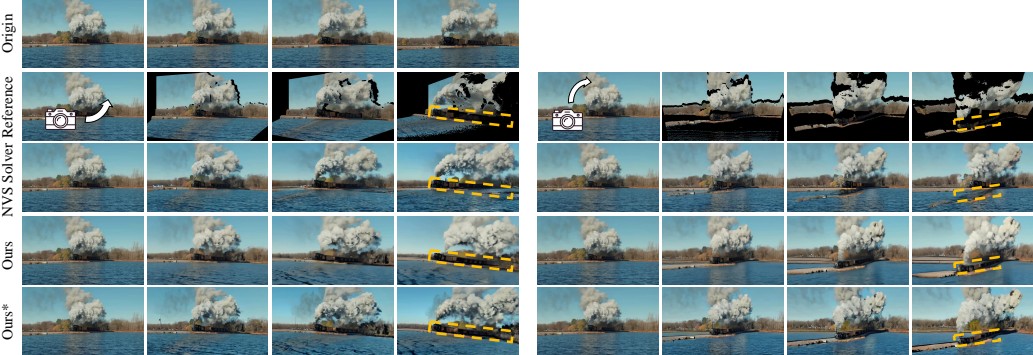

Figure 7: **Qualitative comparisons for camera motion control on videos.** In the second row, we provide video frames after view warping as a reference. Methods like NVS Solver (You et al., 2024) use frame-wise information injection but overlook temporal continuity, leading to inconsistent motion control, especially in frames farther from the first one. In contrast, our approach explicitly models attention across frames, which significantly benefits control precision. We highlight the control precision with yellow boxes, where our method aligns better with the reference. *: we integrate NVS Solver's capability to inject frame-wise information, achieving better video alignment with the original videos.

### 5.3 CAMERA MOTION CONTROL ON VIDEOS

We compare the video synthesis performance of our method with You et al. (2024), who employ a test-time optimization approach. Their method uses view-warped frames as optimization targets, injecting partial frame information into the generation process. However, it optimizes on a per-frame basis, neglecting temporal coherence. As a result, when large view changes occur, their method often struggles to follow the motion accurately and introduces spatial blur. In contrast, our method precisely handles large motions. Notably, the way You et al. (2024) injects frame information is orthogonal to our approach. By combining their optimization technique with our trajectory attention, we achieve higher fidelity in the generated results, as demonstrated in Table 2 and Fig. 7.

### 5.4 VIDEO EDITING

Compared to previous first-frame guided editing methods (Ku et al., 2024; Ouyang et al., 2024), our approach explicitly models motion dynamics as trajectories across frames, enabling better content consistency over large spatial and temporal ranges. As shown in Fig. 8, while other methods struggle to maintain consistency after editing, our method successfully preserves the edited features throughout the entire sequence.

### 5.5 ABLATION ON TRAJECTORY ATTENTION DESIGNS

To validate the effectiveness of our trajectory attention design, we conducted an ablation study, presented in Table 3 . We examined four types of implementations: 1) Directly applying temporal attention to trajectory attention, 2) Integrating trajectory attention into temporal attention with weight fine-tuning, 3) Utilizing an add-on branch for modeling trajectory attention, and 4) Inheriting weights from temporal attention (as illustrated in Fig. 5)

The results in Table 3 indicate that the vanilla adaptation leads to significantly poor motion tracking and video quality, with some outputs exhibiting complete noise (we omit such invalid results during

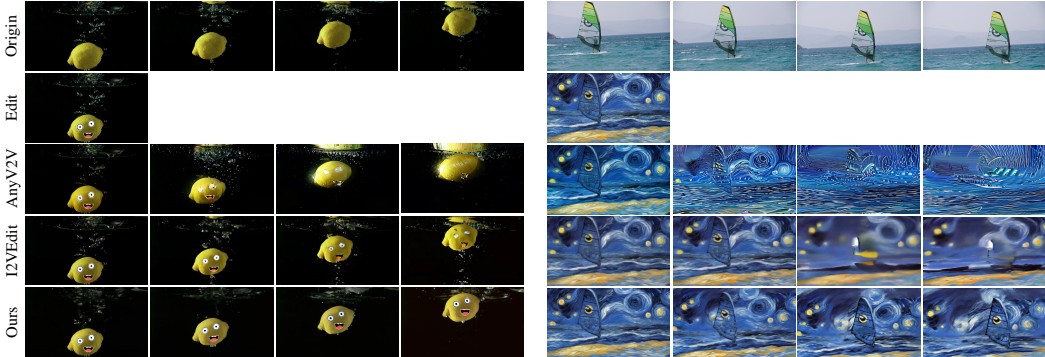

Figure 8: **Results on first-frame guided video editing.** We compare our method with those from Ouyang et al. (2024); Ku et al. (2024). The results show that other methods struggle to maintain consistency after editing. In contrast, our method successfully preserves the edited features across frames, thanks to its ability to model trajectory consistency throughout the video.

Table 3: Ablation on trajectory attention design.

| Methods | ATE (m, ↓) | RPE trans (m, ↓) | RPE Rot (deg, ↓) | FID (↓) |
|---|---|---|---|---|
| Vanilla | 1.7812 | 2.4258 | 13.2141 | 329.6 |
| + Tuning | 0.3147 | 0.3169 | 1.5364 | 139.2 |
| + Add-on Branch | 0.0724 | 0.1274 | 0.3824 | 112.4 |
| + Weight Inheriting | **0.0396** | **0.0232** | **0.1939** | **103.5** |

evaluation, otherwise calculating the statistic results is not feasible.). After fine-tuning the temporal weights, the implementation functions better but remains suboptimal. In contrast, using an add-on branch for trajectory attention markedly improves both motion control precision and video quality. Additionally, inheriting weights from temporal attention facilitates faster convergence and better overall performance compared to simply initializing attention weights randomly.

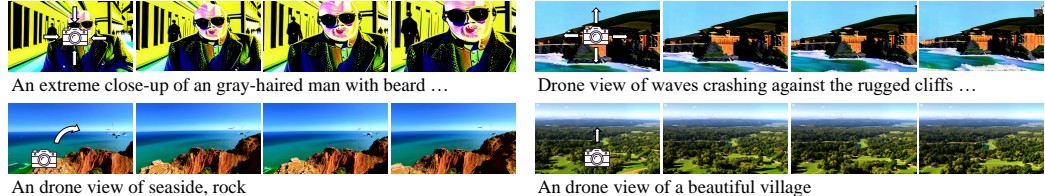

An extreme close-up of an gray-haired man with beard …

Drone view of waves crashing against the rugged cliffs …

An drone view of seaside, rock

An drone view of a beautiful village

Figure 9: **Qualitative results on Open-Sora-Plan.**(Lab & etc., 2024) By incorporating trajectory attention into the 3D attention module, we successfully enable camera motion control.

## 5.6 RESULTS ON FULL ATTENTION MODELS.

Our method also has the potential to support full 3D attention using a similar pipeline as shown in Fig. 3 and Fig. 5, with the key difference being the application of trajectory attention to the 3D attention module instead of the temporal attention. As demonstrated in Fig. 9, this enables diverse camera motion control in the generated results. For detailed implementation, please refer to the supplementary materials.

## 6 CONCLUSION

In conclusion, we introduced trajectory attention, a novel approach for fine-grained camera motion control in video generation. Our method, which models trajectory attention as an auxiliary branch alongside temporal attention, demonstrates significant improvements in precision and long-range consistency. Experiments show its effectiveness in camera motion control for both images and videos while maintaining high-quality generation. The approach's extensibility to other video motion control tasks, such as first-frame-guided video editing, highlights its potential impact on the broader field of video generation and editing.

**Acknowledgements.** This research is supported by MOE AcRF Tier 1 (RG97/23) and NTU SUG-NAP. This research is also supported under the RIE2020 Industry Alignment Fund – Industry Collaboration Projects (IAF-ICP) Funding Initiative, as well as cash and in-kind contribution from the industry partner(s).

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
