# OpenReview forum: "Trajectory attention for fine-grained video motion control"
_ICLR.cc/2025/Conference — ICLR 2025 Poster_

### Official Review · Reviewer_KvZ7 · 2024-10-16

**Soundness:** 3
**Presentation:** 3
**Contribution:** 3
**Rating:** 8
**Confidence:** 5

**Summary:**

This paper introduces Trajectory Attention, an innovative method for fine-grained camera motion control that attends to available pixel trajectories. The authors identify conflicts between the original temporal attention modules in diffusion models and supplementary trajectory-conditioned temporal modules. To resolve these conflicts, the paper employs optical-flow data to define trajectories, samples the most correlated points along them, and applies a copy attention mechanism to enhance trajectory precision. The original temporal module is retained for consistency. Comprehensive experiments on camera motion control for both images and videos demonstrate significant improvements in precision and long-range consistency without compromising high-quality generation. Furthermore, the approach is shown to be extensible to other video motion control tasks, including first-frame-guided video editing, where it maintains content consistency over extensive spatial and temporal dimensions

**Strengths:**

- The Trajectory Attention module is intuitive and offers flexibility in capturing temporal correlations in camera motion. This innovative approach effectively addresses the challenges associated with fine-grained control of camera motion.

- The experiments on camera motion control for both images and videos are impressive. They demonstrate significant improvements in precision and long-range consistency, all while maintaining high-quality generation. These results underscore the effectiveness of the proposed method in handling complex camera motion scenarios.

- The paper effectively shows that the approach can be extended to other video motion control tasks. For instance, in first-frame-guided video editing, the method excels at maintaining content consistency over large spatial and temporal ranges. This versatility is a testament to the robustness and general applicability of the Trajectory Attention framework.

**Weaknesses:**

- The paper raises concerns about object dynamics in the image-to-video case presented in the supplementary material. The examples, such as the dog and the cat, lack additional motion, which could be a limitation. It would be beneficial to see how objects with more complex dynamics are handled by the method.

- There is a concern regarding the generalization of camera pose. In the Image-to-Video (first-frame) scenario, the trajectory module is trained with optical-flow data from only 10K video clips. It's unclear how the method would perform under challenging motions, such as clockwise rotation, high-speed zooming in and out, or 360-degree rotations like those seen in NVS-Solver GitHub. In these extreme trajectories, points visible in the first frame may become invisible, potentially leading to anti-aliasing issues. Additional results or a discussion of the necessary limitations would aid in a more comprehensive assessment of the proposed method.

**Questions:**

- How can one obtain or customize the appropriate intrinsic and extrinsic parameters when performing trajectory extraction for a single image or video? Does the camera always need to be directed at the center of the image?

- Is it necessary to adjust the camera's intrinsic and extrinsic parameters based on the depth information available?

---

> ### Author Response · Authors · 2024-11-22
>
> We sincerely appreciate your thorough review and feedback! Please find our detailed responses to your observations and suggestions below.
>
> ---
>
> **W1: Handling complex object dynamics in Image-to-Video cases**
> > The paper raises concerns about object dynamics in the image-to-video case presented in the supplementary material. The examples, such as the dog and the cat, lack additional motion, which could be a limitation. It would be beneficial to see how objects with more complex dynamics are handled by the method.
>
> The outcome dynamics depend on the region where trajectory control is applied and the sparsity of the control signal. By default, dense control is applied to all pixels, resulting in pixel-wise alignment. In contrast, when using sparser trajectories, the results exhibit greater variability, as illustrated in Fig. 8 (c) in the supplementary material (please also see the attached videos for better visualization). However, this approach involves a trade-off between control precision and the generated dynamics.
>
> ---
>
> **W2: Generalization of camera pose in Image-to-Video scenarios**
> > There is a concern regarding the generalization of camera pose. In the Image-to-Video (first-frame) scenario, the trajectory module is trained with optical-flow data from only 10K video clips. It's unclear how the method would perform under challenging motions, such as clockwise rotation, high-speed zooming in and out, or 360-degree rotations like those seen in NVS-Solver GitHub. In these extreme trajectories, points visible in the first frame may become invisible, potentially leading to anti-aliasing issues. Additional results or a discussion of the necessary limitations would aid in a more comprehensive assessment of the proposed method.
>
> Since our method does not rely on training with camera-annotated datasets, it can naturally generalize to various camera poses. As demonstrated in Fig. 6 of the supplementary materials, our approach effectively handles challenging scenarios such as high-speed zooming and clockwise rotation. However, achieving 360-degree rotations with 3D cycle consistency poses challenges for our method. Implementing 360-degree rotations would require additional design considerations, such as using both the starting and ending frames to perform interpolation tasks, similar to those in NVS-Solver [5]. We have also introduced necessary constraints to address these challenges (please see the limitation discussion in the supplementary material for details).
>
> ---
>
> **Q1: Customization of intrinsic and extrinsic parameters**
> > How can one obtain or customize the appropriate intrinsic and extrinsic parameters when performing trajectory extraction for a single image or video? Does the camera always need to be directed at the center of the image?
>
> Since we cannot precisely estimate the intrinsic and extrinsic parameters from a single image, we use predefined intrinsic parameters and some hyperparameters for extrinsic parameters. From our observations, these predefined parameters with statistics can effectively generate reasonable results. We can also adjust them accordingly.
>
> As our approach is independent of specific camera settings and relies solely on generated trajectories, the camera's direction can be adjusted freely. For instance, in Fig. 8 (c) in the supplementary material, the camera is oriented towards the right side of the scene.
>
> ---
>
> **Q2: Dependency on depth information for parameter adjustments**
> > Is it necessary to adjust the camera's intrinsic and extrinsic parameters based on the depth information available?
>
> From our observations, these predefined parameters with statistics can effectively generate reasonable results. We can also adjust them accordingly.

---

> ### Author Response · Authors · 2024-11-25
> **Discussion Period Ending Soon**
>
> Dear Reviewer,
>
> Thank you for your valuable feedback and insightful comments throughout the discussion phase.
>
> As the discussion period concludes on November 26th, we kindly ask if our responses have effectively addressed your questions. Please don’t hesitate to reach out with any additional questions, concerns, or requests for clarification.
>
> Warm regards,
>
> The Authors

---

> > ### Comment · Reviewer_KvZ7 · 2024-11-27
> >
> > Thanks for the author's reply, the paper still lacks some discussion of limitations and shows some failure examples to avoid cherry-picking. Besides, as your answers to Q2, showing some cases for adjusting intrinsic parameters will differ your works to others.

---

> > > ### Author Response · Authors · 2024-11-27
> > >
> > > Thank you for your valuable suggestions.
> > >
> > > We have incorporated additional failure cases highlighting the limitations in Fig. 11 of the supplementary material. Furthermore, we have included examples showing the effects of adjusting intrinsic parameters in Fig. 12 of the supplementary material.
> > >
> > > We hope these results could address your concerns effectively.

---

> > > > ### Comment · Reviewer_KvZ7 · 2024-11-29
> > > >
> > > > Great, How will you address the high-speed camera motion and complex motion,  will some mask for the uncertain area help? Besides, I raise my score.

---

> > > > > ### Author Response · Authors · 2024-11-29
> > > > >
> > > > > Thank you for your valuable feedback. We are delighted to hear that you are satisfied with our response.
> > > > >
> > > > > When it comes to high-speed or complex motion, since our method highly depends on the generation ability of the base models, we believe the key challenge lies in advancing the generative capabilities of foundational models. Currently, even state-of-the-art video diffusion models, such as Kling, face significant limitations in generating realistic results for complex motion scenarios. We acknowledge this as an open challenge and leave it for future exploration.

---

### Official Review · Reviewer_1VgT · 2024-11-02

**Soundness:** 3
**Presentation:** 2
**Contribution:** 3
**Rating:** 8
**Confidence:** 4

**Summary:**

The paper focuses on fine-grained camera motion control in video generation. It has the following contributions:
1. Trajectory Attention Mechanism: Proposes a novel trajectory attention branch alongside the original temporal attention branch. It models attention along available pixel trajectories for camera motion control.
2. Improved Performance: Demonstrates significant improvements in precision and long-range consistency for camera motion control in both images and videos while maintaining high-quality generation.
3. Extension to Other Tasks: Shows that the approach can be extended to other video motion control tasks, such as first-frame-guided video editing, where it excels in maintaining content consistency over large spatial and temporal ranges.

**Strengths:**

1. Originality: The paper demonstrates originality in its approach to video motion control. The concept of trajectory attention, modeled as an auxiliary branch to traditional temporal attention, is a novel way to incorporate pixel trajectories for fine-grained camera motion control. This approach differs from existing methods that either rely on high-level constraints or neglect temporal correlations.
2. Quality: The experimental setup is comprehensive, using a large-scale dataset and multiple evaluation metrics. The results are presented in a clear and organized manner, with both quantitative comparisons and qualitative visualizations. The ablation study further validates the effectiveness of the proposed components, indicating a high level of quality in the research design and execution.
3. Significance: The significance of the paper lies in its potential impact on the field of video generation and motion control. The proposed method shows improved performance in camera motion control for both images and videos, which is crucial for creating high-quality and customized visual content.

**Weaknesses:**

1. The paper does discuss trajectory extraction for different tasks such as camera motion control on images and videos, and video editing. However, the description of the extraction process could be more detailed and clear. For example, in Algorithm 3 for trajectory extraction from a single image, some steps might require further clarification for a reader who is not familiar with the underlying concepts. The estimation of the depth map and the rendering of views are steps that could be explained in more detail, including the methods and algorithms used. Similarly, in Algorithm 4 for video trajectory extraction, the point trajectory estimation and the combination with camera motion could be more clearly described.
2. While the proposed trajectory attention method shows promising results in the presented experiments, there is a lack of exploration of more complex scenarios. For example, in real-world video data, there may be occlusions, rapid camera movements, or multiple moving objects, and it is not clear how the method would perform in such situations.
3. The comparison with existing methods, although extensive to some extent, could be more comprehensive. There are many other techniques in the field of video motion control that were not included in the comparison, and it is possible that some of these methods may have unique features or advantages that could have provided a more nuanced understanding of the proposed method's superiority.

**Questions:**

According to the weaknesses, you can take the suggestions below to make your paper more convincing:
1. In Algorithm 3, please elaborate on which depth estimation method you take in step 1 and how you render a set of views $I_{r}$ and get the translation of pixels $T$ in step 2. In Algorithm 4, please elaborate on which point trajectory estimation method you take in step 2. Meanwhile, could you provide the visual results of the trajectory extraction from a single image and a video to demonstrate the correctness of Algorithms 3 and 4?
2. Provide results of your method on videos with occlusions, rapid camera movements, and multiple moving objects, respectively.
3. Provide a comparison with more related works, such as [MotionBooth](https://arxiv.org/abs/2406.17758) and [CamTrol](https://arxiv.org/abs/2406.10126). More comparisons to concurrent work are also encouraged but not mandatory.

If the authors could solve my problems, I would raise the score.

---

> ### Author Response · Authors · 2024-11-22
>
> Thank you for your detailed comments and suggestions! We have reviewed them carefully and provided explanations below to clarify the points of concern.
>
> ---
>
>
> **W1&Q1: Clarification of trajectory extraction processes**
> > The paper does discuss trajectory extraction for different tasks such as camera motion control on images and videos, and video editing. However, the description of the extraction process could be more detailed and clear. For example, in Algorithm 3 for trajectory extraction from a single image, some steps might require further clarification for a reader who is not familiar with the underlying concepts. The estimation of the depth map and the rendering of views are steps that could be explained in more detail, including the methods and algorithms used. Similarly, in Algorithm 4 for video trajectory extraction, the point trajectory estimation and the combination with camera motion could be more clearly described.
>
> Thank you for your feedback. We have included additional details in the supplementary material for clarification. For more comprehensive explanations regarding the estimation methods we use, clarification of certain concepts, and visualizations, please refer to Sec. A.3 and A.4 in the supplementary material. We will continue to address any remaining points if further clarification is required.
>
> ---
>
> **W2&Q2: Handling complex real-world scenarios**
> > While the proposed trajectory attention method shows promising results in the presented experiments, there is a lack of exploration of more complex scenarios. For example, in real-world video data, there may be occlusions, rapid camera movements, or multiple moving objects, and it is not clear how the method would perform in such situations.
>
> Thank you for your suggestions. We have included more examples of complex scenarios. As illustrated in Fig. 6 of the supplementary materials, our method effectively handles challenging cases such as occlusions, rapid camera movements (e.g., zooming in and out), and multiple moving objects. Additionally, we have expanded the discussion in the limitations section to provide a more comprehensive understanding of our approach.
>
> ---
>
> **W3&Q3: Comprehensive comparison with existing methods**
> > The comparison with existing methods, although extensive to some extent, could be more comprehensive. There are many other techniques in the field of video motion control that were not included in the comparison, and it is possible that some of these methods may have unique features or advantages that could have provided a more nuanced understanding of the proposed method's superiority.
>
> Thank you for your suggestions. We have conducted comparisons with the most relevant open-source methods (MotionCtrl [3], CameraCtrl [4], NVS Solver [5], Motion-I2V [6], anyV2V [7], I2VEdit [8]) in our experiments. For other related methods, we have revised the paper to include discussions. For example, while MotionBooth[12] offers camera motion control, its effectiveness is demonstrated only for simple pan motions. CamTrol [13] enables camera control by rendering incomplete warped views followed by re-denoising, which may become less effective when handling large incomplete regions. We were unable to include direct comparisons with it because we can not reach their code currently. For further detailed discussions, please refer to the revised paper, particularly the related work section.

---

> > ### Comment · Reviewer_1VgT · 2024-11-26
> >
> > Sorry for the late response, I still have some questions:
> > 1. How to understand the description of "Render a set of views {$I_r$} using D, K, and E"? It seems that there is no explanation in your supplementary materials.
> > 2. Can you provide visualization results for video editing with multiple objects that belong to distinct classes?

---

> > > ### Author Response · Authors · 2024-11-27
> > >
> > > Thank you for your response!
> > >
> > > We recognize that the description of "Render ..." might have been somewhat misleading. The actual purpose of this line in the algorithm is to compute pixel translations, a process that mimics rendering. To clarify, we have removed this description. For details on how pixel translations are obtained, please refer to Algorithm 1 in the supplementary materials.
> > >
> > > Furthermore, we have provided an example featuring a woman and a dog in Fig. 6(d). Our method demonstrates the ability to capture subtle motions, such as a slight head turn, highlighting its fine-grained capabilities. It is also important to emphasize that our approach is inherently class-agnostic, relying solely on motion flow, allowing it to handle such scenarios with ease.
> > >
> > > We are happy to provide further clarifications if needed.

---

> > > > ### Comment · Reviewer_1VgT · 2024-11-27
> > > >
> > > > Thanks for your response. All my concerns have been addressed and I can raise my score.

---

> > > > > ### Author Response · Authors · 2024-11-28
> > > > >
> > > > > Thank you for your thoughtful feedback and for taking the time to review our response. We're glad to hear that your concerns have been addressed. Your support and insights are greatly appreciated!

---

> ### Author Response · Authors · 2024-11-25
> **Discussion Period Ending Soon**
>
> Dear Reviewer,
>
> Thank you for your valuable feedback and insightful comments throughout the discussion phase.
>
> As the discussion period concludes on November 26th, we kindly ask if our responses have effectively addressed your questions. Please don’t hesitate to reach out with any additional questions, concerns, or requests for clarification.
>
> Warm regards,
>
> The Authors

---

### Official Review · Reviewer_EFMe · 2024-11-02

**Soundness:** 2
**Presentation:** 2
**Contribution:** 3
**Rating:** 6
**Confidence:** 3

**Summary:**

This paper proposes injecting a new attention layer along the trajectory into the model to support camera motion control in video generation. During training, optical flow is used as the trajectory, and the new attention operation is performed only along this trajectory. The trained model achieves good results in camera control for image-to-video and video-to-video tasks.

**Strengths:**

- Metric-wise, it seems the model achieves better camera control.
- The model can be used for first-edited-frame + original-video-guided editing, though how this is achieved is not very clear.

**Weaknesses:**

1) Figure-1 is confusing. It takes some time to understand the input and output of each task. It would be better to reorganize this figure to make it clearer. Each task could be separated into a small sub-figure with a clear indication of the input and output.

2) In Figure-3, it’s unclear what the model’s input is in two scenarios: (1) when you have multiple frames as input, i.e., ‘camera motion control on videos’ in Figure-1, and (2) when you have multiple frames plus edited frames as input, i.e., ‘first-frame-guided video editing’ in Figure-1.

3) The trajectory attention mechanism operates only in 2D space, making it challenging to distinguish motion orthogonal to the image plane—for example, whether an centered object is moving towards or away from the camera. In such cases, the coordinates remain the same across frames. Could this be a limitation of this method?

**Questions:**

1) How do you ensure that when attention is applied along the trajectory, the generated pixel also follows the trajectory? Have you observed any cases where control fails?

2) In Algorithm 3, are you feeding \{I_r\} to the model in any particular format? The same question applies for Algorithm 4 with \{V_r\}.

3) Is the comparison to other work (motion control/camera control) fair? They are trained on different datasets, and they may have some issue generalizing to the evaluation dataset used here. How did you select the evaluation set? Were you able to evaluate on the test set of other papers?

4) In training, optical flow is used as a trajectory, but in inference, the model takes the camera trajectory as input. Could this cause a mismatch between training and inference? Why not use the camera trajectory as guidance during training as well?

---

> ### Author Response · Authors · 2024-11-22
>
> We’re grateful for your constructive suggestions! Below, we’ve outlined our responses to address each of your concerns.
>
> ---
>
> **W1: Figure-1 clarity improvements**
> > Figure-1 is confusing. It takes some time to understand the input and output of each task. It would be better to reorganize this figure to make it clearer.
>
> Thank you for your suggestions. We have revised Figure 1 accordingly. To enhance clarity, we have used distinct colors to differentiate the reference contents, inputs, and outputs.
>
> ---
>
> **W2: Input scenarios in Figure-3**
> > In Figure-3, it’s unclear what the model’s input is in two scenarios.
>
> The primary purpose of Figure 3 is to illustrate the trajectory-conditioned generation process, which is general to the tasks discussed in the paper. The main distinction between these tasks lies in the trajectory extraction process, detailed in Sec. 4. However, we acknowledge that it is better to illustrate all these scenarios. Due to page limitations, these additional demonstrations have been included in the supplementary material. Please see Section A.3 for more explanations, as well as Fig. 4 for the visualization of the input of these two scenarios.
>
> ---
>
> **W3: Motion orthogonal to the image plane**
> > The trajectory attention mechanism operates only in 2D space, making it challenging to distinguish motion orthogonal to the image plane—for example, whether a centered object is moving towards or away from the camera. The coordinates remain the same across frames. Could this be a limitation of this method?
>
> Motion can be modeled whenever there are pixel shifts within the image space. For example, when an object moves toward the camera, it occupies more pixels due to perspective projection. To further support this concept, we have included additional examples in the supplementary material for verification. Please refer to Fig. 6 for examples of zooming-in and zooming-out scenarios.
>
> ---
>
> **Q1: Trajectory adherence in generated pixels**
> > How do you ensure that when attention is applied along the trajectory, the generated pixel also follows the trajectory? Have you observed any cases where control fails?
>
> The generated pixel can follow the trajectory because 1) the trajectory attention is trained to generate consistent results, and 2) the design of trajectory attention has a strong inductive bias, i.e., the attention has a specific goal with little ambiguity, making it easy to train and generalize. Because of this, we rarely see any failure cases. The control performance would degrade only when the motion trajectories are extremely sparse, e.g., below 1/32 of the original trajectories (Fig. 8 (a) in the supplementary material).
>
> **Q2: Explanation of {I_r} and {V_r} usage in Algorithms 3 and 4**
> > In Algorithm 3, are you feeding {I_r} to the model in any particular format? The same question applies for Algorithm 4 with {V_r}.
>
> Our input consists solely of the first frame and the extracted trajectory. The {I_r} and {V_r} are used for trajectory extraction. For more details, please refer to Fig. 4 in the supplementary material.
>
> ---
>
> **Q3: Fairness in evaluation comparisons**
> > Is the comparison to other work (motion control/camera control) fair? They are trained on different datasets, and they may have some issue generalizing to the evaluation dataset used here. How did you select the evaluation set? Were you able to evaluate on the test set of other papers?
>
> We have ensured a reasonably fair comparison. For the evaluation dataset, since most related works do not provide their datasets, we selected data from publicly available sources and datasets (e.g., DAVIS [11]) that are distinct from our training dataset. For the training dataset, MotionCtrl [3] and CameraCtrl [4] require specially annotated camera parameters, whereas our method only requires video datasets without such annotations. (Note that our method is also not sensitive to the dataset, as shown in Table 1 in the supplementary material.)
>
> ---
>
> **Q4: Training and inference trajectory consistency**
> > In training, optical flow is used as a trajectory, but in inference, the model takes the camera trajectory as input. Could this cause a mismatch between training and inference? Why not use the camera trajectory as guidance during training as well?
>
> The core idea of our work is to use trajectories for motion control. Camera trajectory is handled by first converting to pixel trajectories, and then seamlessly processed with our framework. As our method is still working with pixel trajectory, there is no mismatch between training and inference.
>
> While MotionCtrl [3] and CameraCtrl [4] are specifically designed for camera control and use camera trajectory as a direct condition, we demonstrate in the paper that such high-level conditioning does not achieve precise control. Our trajectory attention, due to its strong inductive bias, is easier to train and to learn more precise control.

---

> ### Author Response · Authors · 2024-11-25
> **Discussion Period Ending Soon**
>
> Dear Reviewer,
>
> Thank you for your valuable feedback and insightful comments throughout the discussion phase.
>
> As the discussion period concludes on November 26th, we kindly ask if our responses have effectively addressed your questions. Please don’t hesitate to reach out with any additional questions, concerns, or requests for clarification.
>
> Warm regards,
>
> The Authors

---

> > ### Comment · Reviewer_EFMe · 2024-11-27
> > **clarity improvements**
> >
> > Thanks for improving figure-1 and adding figure-4.
> > * In figure-4, are you using the warpped input in any form? from the arrow it seems they are dropped?
> > * In figure-1, there is still some black boundary at the bottom right corner. Are they artifact of the generation?
> > * How do you feed the reference frames of figure-1 row-3, yellow box, into the model? I feel this is not clear in figure-4.

---

> > > ### Comment · Reviewer_EFMe · 2024-11-27
> > >
> > > As most of my concern as addressed, I raise my score. But the Figure-4 still need more improvement, from the figure, it's not clear how each component of the framework looks like.

---

> > > > ### Author Response · Authors · 2024-11-27
> > > >
> > > > Thank you for acknowledging our response and efforts.
> > > >
> > > > In the revised paper, we have provided additional descriptions for Fig. 4 to enhance clarity.
> > > >
> > > > Specifically, for all tasks, the inputs to the network consist of the first frame and the extracted trajectories. The usage of the first frame and trajectories remains consistent with Fig. 3 in the main paper.
> > > >
> > > > The wrapped input is solely for visualization purposes and is not utilized in the pipeline. Similarly, the reference frames are employed only to extract point trajectories and are not involved in the pipeline.
> > > >
> > > > Regarding the black boundary observed in row 2 of Fig. 1, it is not an artifact. These black shadows are present in the original videos and are faithfully reflected in the generated results.

---

### Official Review · Reviewer_v5kt · 2024-11-03

**Soundness:** 3
**Presentation:** 3
**Contribution:** 3
**Rating:** 6
**Confidence:** 4

**Summary:**

The paper introduces Trajectory Attention, a novel approach designed to enhance fine-grained motion control in video generation, particularly focusing on precise camera motion control within video diffusion models. Traditional methods often struggle with imprecise outputs and neglect temporal correlations, leading to inconsistencies in generated videos. This work addresses these challenges by explicitly modeling trajectory attention as an auxiliary branch alongside the standard temporal attention mechanism. By modeling trajectory attention as an auxiliary branch alongside the standard temporal attention, the method explicitly injects available pixel trajectory information into the video generation process. This design allows the temporal attention to focus on motion synthesis and short-range dynamics, while the trajectory attention ensures long-range consistency along specified paths. The approach efficiently integrates trajectory information without modifying the original model parameters and supports sparse trajectories, meaning it can handle partial trajectory data. Experiments demonstrate that this method significantly improves motion control precision and video quality across various tasks, including camera motion control on images and videos, as well as first-frame-guided video editing.

**Strengths:**

1. The paper introduces a novel concept of Trajectory Attention for fine-grained motion control in video generation. This auxiliary attention mechanism enhances the existing temporal attention in video diffusion models by explicitly incorporating trajectory information, which is a significant advancement in the field.
2. By modeling trajectory attention as an auxiliary branch that works alongside the original temporal attention, the approach allows for seamless integration without modifying the original model parameters. This design choice is both practical and efficient, leveraging pre-trained models and enabling efficient fine-tuning.
3. The proposed method demonstrates significant improvements in motion control precision and long-range consistency over existing methods. The experimental results, including quantitative metrics like Absolute Trajectory Error (ATE) and Relative Pose Error (RPE), validate the effectiveness of the approach.
4. The paper includes thorough experiments and ablation studies that not only demonstrate the superior performance of the proposed method but also validate the design choices. This strengthens the credibility of the findings and provides valuable insights into the method's effectiveness.

**Weaknesses:**

1. The method is primarily designed for video diffusion models that use decomposed spatial-temporal attention. It is less clear how well the approach generalizes to models with integrated spatial-temporal attention (e.g. 3D DiTs) or other architectures. Expanding the evaluation to include such models would strengthen the contribution.
2. The paper compares the proposed method with a limited set of existing approaches. Including discussions with more recent or state-of-the-art methods, especially those that have emerged concurrently, would provide a more comprehensive evaluation of the method's relative performance. For example, Collaborative Video Diffusion [1] uses epipolar attention to align contents of different camera trajectories, and Camco [2] also uses epipolar, but to enhance the 3D consistency of generated contents.
3. The experimental evaluations are primarily conducted on the MiraData dataset. While this dataset may offer certain advantages, relying on a single dataset limits the ability to generalize the findings. Evaluating the method on additional, diverse datasets would strengthen the claims about its general applicability.
4. While the method supports sparse trajectories, the paper does not extensively explore how it performs when the trajectory information is highly sparse, incomplete, or noisy. Real-world applications often involve imperfect data, so robustness to such conditions is important. Going back to my point 2, this is especially concerning since the model is trained on MiraData, which mostly consists of synthetic videos.

[1] Kuang et al. Collaborative Video Diffusion: Consistent Multi-video Generation with Camera Control, in NeurIPS, 2024.

[2] Xu et al. CamCo: Camera-Controllable 3D-Consistent Image-to-Video Generation, in arXiv, 2024.

**Questions:**

N/A

---

> ### Author Response · Authors · 2024-11-22
>
> Thank you for your insightful feedback! We have addressed your concerns and provided detailed responses to each point below.
>
> ---
>
> **W1: Expanding to other architectures**
> > The method is primarily designed for video diffusion models that use decomposed spatial-temporal attention. It is less clear how well the approach generalizes to models with integrated spatial-temporal attention (e.g., 3D DiTs) or other architectures. Expanding the evaluation to include such models would strengthen the contribution.
>
> Our method can be seamlessly extended to DiT [10], as the key insight of modifying attention across frames remains applicable. We present qualitative results in Fig. 9 and provide a detailed explanation of the 3D DiT approach in the supplementary material (Section A.2) due to space constraints.
>
> ---
>
> **W2: Comparisons with recent or state-of-the-art methods**
> > The paper compares the proposed method with a limited set of existing approaches. Including discussions with more recent or state-of-the-art methods, especially those that have emerged concurrently, would provide a more comprehensive evaluation of the method's relative performance. For example, Collaborative Video Diffusion uses epipolar attention to align contents of different camera trajectories, and Camco also uses epipolar, but to enhance the 3D consistency of generated contents.
>
> Thank you for your suggestions. We have conducted comparisons with the most relevant open-source methods in our experiments (i.e., MotionCtrl [3], CameraCtrl [4], NVS Solver[5], Motion-I2V [6], anyV2V [7], I2VEdit [8]). Additionally, we have revised the paper to include discussions on other concurrent methods. For instance, Collaborative Video Diffusion [1] introduces a collaborative structure with epipolar attention for consistent camera-controlled generation, while Camco [2] also leverages the epipolar constraint for generation. However, the epipolar constraint is, in fact, a weaker variant of trajectory attention. Moreover, due to the current unavailability of their code, we could not include direct comparisons. For more in-depth discussions, please refer to the revised paper, particularly the related work section.
>
> ---
>
> **W3: Dataset in the experimental evaluations**
> > The experimental evaluations are primarily conducted on the MiraData [9] dataset. While this dataset may offer certain advantages, relying on a single dataset limits the ability to generalize the findings. Evaluating the method on additional, diverse datasets would strengthen the claims about its general applicability.
>
> Thanks to the strong inductive bias of our trajectory attention design, our method is data efficient and can generalize well even with a single dataset. To validate this claim, we have included a new table (Table 1) in the supplementary material, which shows that our approach is not sensitive to the dataset size or the training domains.
>
> ---
>
> **W4: Robustness to sparse, incomplete, or noisy trajectory information**
> > While the method supports sparse trajectories, the paper does not extensively explore how it performs when the trajectory information is highly sparse, incomplete, or noisy. Real-world applications often involve imperfect data, so robustness to such conditions is important. Going back to my point 2, this is especially concerning since the model is trained on MiraData, which mostly consists of synthetic videos.
>
> MiraData [9] actually incorporates lots of real-world data, with the training set primarily consisting of such samples. As illustrated in Fig. 10 of the supplementary material, the estimated optical flow used as training input is notably sparse and incomplete. This characteristic contributes to the robustness of our methods. Nonetheless, we acknowledge that our methods have limitations when handling extremely sparse trajectories (see Fig. 8 in the supplementary material), suggesting an intriguing direction for future research.

---

> ### Author Response · Authors · 2024-11-25
> **Discussion Period Ending Soon**
>
> Dear Reviewer,
>
> Thank you for your valuable feedback and insightful comments throughout the discussion phase.
>
> As the discussion period concludes on November 26th, we kindly ask if our responses have effectively addressed your questions. Please don’t hesitate to reach out with any additional questions, concerns, or requests for clarification.
>
> Warm regards,
>
> The Authors

---

### Official Review · Reviewer_CnyS · 2024-11-04

**Soundness:** 3
**Presentation:** 3
**Contribution:** 3
**Rating:** 6
**Confidence:** 4

**Summary:**

The paper introduces a novel approach called Trajectory Attention for fine-grained video motion control, particularly aiming to enhance camera motion control in video generation tasks. By modeling trajectory attention as an auxiliary branch alongside traditional temporal attention, the method leverages available pixel trajectories to inject precise motion information into the video generation process. This design allows the original temporal attention and the trajectory attention to work synergistically. The proposed method demonstrates strong adaptability, e.g., being transferable to architectures like DiT. Experiments across various tasks show significant improvements in control precision and content consistency while maintaining high-quality generation. Extensive ablation studies validate the effectiveness of each module.

**Strengths:**

1. The proposed method is lightweight, requiring low training costs, making it practical and efficient for real-world applications without the need for extensive computational resources.
2. The method demonstrates strong transferability, showing effectiveness with different architectures such as DiT.
3. The paper conducts thorough exploration at application level, showcasing the method's effectiveness in multiple tasks, including camera motion control and video editing. Abalation studies are sufficient.

**Weaknesses:**

1. The method heavily relies on dense optical flow information, as shown in Figure 3 of the supplementary material. This dependency can significantly increase inference time due to the computational cost of processing dense optical flow, especially in real-time applications.
2. The reliance on dense optical flow makes it challenging to adapt the method to user inputs of sparse trajectories. As noted in DragNUWA, it's difficult for users to input precise trajectories at key points in practical applications, leading to a gap between training and inference. This limitation reduces the method's practicality in scenarios where only sparse motion cues are available.
3. In line 158, H and W represent the dimensions of the latent features, but in Algorithm 3, H and W are used for image dimensions, which is confusing.
4. Some examples in Fig.6 and Fig.9 are not significant, like the second example in Fig.6.

**Questions:**

I suggest to review and correct the mathematical formulations and notation to enhance the paper's clarity and reliability.

---

> ### Author Response · Authors · 2024-11-22
>
> Thank you for your thoughtful feedback! We have provided clarifications and responses to your concerns below.
>
> ---
>
> **W1: Explanation of dense optical flow dependency**
> > The method heavily relies on dense optical flow information, as shown in Figure 3 of the supplementary material. This dependency can significantly increase inference time due to the computational cost of processing dense optical flow, especially in real-time applications.
>
> Processing optical flow does not significantly increase the overall time cost. For instance, generating or predicting dense optical flow for a video with a resolution of 1024×576 and 25 frames takes approximately 20 seconds, accounting for around 20% of the total inference time. This overhead is reasonable for video generation tasks. Also, our methods support relatively sparse trajectories, as shown in Fig. 8 in the supplementary material.
>
> ---
>
> **W2: Explanation of challenges in adapting to sparse trajectories**
> > The reliance on dense optical flow makes it challenging to adapt the method to user inputs of sparse trajectories. As noted in DragNUWA, it's difficult for users to input precise trajectories at key points in practical applications, leading to a gap between training and inference. This limitation reduces the method's practicality in scenarios where only sparse motion cues are available.
>
> Although our experiments primarily leverage dense optical flow, this approach also shows promise for sparse scenarios (as detailed in Section A.8 of the supplementary material). However, we acknowledge that our current methods are less effective at handling highly sparse trajectories. Our techniques are designed to provide a general and robust framework for utilizing available trajectories in motion control, as demonstrated in applications such as camera motion control and video editing. Developing user-friendly sparse trajectory designs, however, remains an exciting avenue for exploration.
>
> ---
>
> **W3: Clarification of H and W usage in the text and Algorithm 3**
> > In line 158, H and W represent the dimensions of the latent features, but in Algorithm 3, H and W are used for image dimensions, which is confusing.
>
> Thank you for your feedback. We have addressed this in the revised rebuttal version.
>
> ---
>
> **W4: Significance of examples in Fig. 6 and Fig. 9**
> > Some examples in Fig. 6 and Fig. 9 are not significant, like the second example in Fig. 6.
>
> Pictures may not always effectively highlight the differences. We recommend viewing the videos included in the supplementary material, where we have also added a video corresponding to Fig. 9.

---

> ### Author Response · Authors · 2024-11-25
> **Discussion Period Ending Soon**
>
> Dear Reviewer,
>
> Thank you for your valuable feedback and insightful comments throughout the discussion phase.
>
> As the discussion period concludes on November 26th, we kindly ask if our responses have effectively addressed your questions. Please don’t hesitate to reach out with any additional questions, concerns, or requests for clarification.
>
> Warm regards,
>
> The Authors

---

### Author Response · Authors · 2024-11-22
**Global response by authors**

We sincerely thank the reviewers for their thoughtful, insightful, and constructive feedback. We are delighted that the originality of our approach to video motion control has been recognized (Reviewers v5kt, KvZ7, 1VgT), and we appreciate the acknowledgment of the technical soundness of our methodology (Reviewers CnyS, v5kt, 1VgT, KvZ7). We are also grateful for the recognition of our method's flexibility and its potential for diverse applications (Reviewers CnyS, v5kt, EFMe, 1VgT, KvZ7), as well as the effectiveness of our trajectory-based attention mechanism (Reviewers CnyS, v5kt, EFMe, 1VgT, KvZ7).

We have carefully addressed each of your comments and provided detailed responses in the attached supplementary materials, along with specific clarifications and discussions below.

Thank you again for your valuable feedback. We look forward to your continued insights and hope that our revisions and explanations meet your expectations.

---

Reference: Due to the character limit for the rebuttals, we've placed the references for all rebuttals below.

[1] Kuang Z, Cai S, He H, et al. Collaborative Video Diffusion: Consistent Multi-video Generation with Camera Control[J]. arXiv preprint arXiv:2405.17414, 2024.

[2] Xu D, Nie W, Liu C, et al. CamCo: Camera-Controllable 3D-Consistent Image-to-Video Generation[J]. arXiv preprint arXiv:2406.02509, 2024.

[3] Wang Z, Yuan Z, Wang X, et al. Motionctrl: A unified and flexible motion controller for video generation[C]//ACM SIGGRAPH 2024 Conference Papers. 2024: 1-11.

[4] He H, Xu Y, Guo Y, et al. Cameractrl: Enabling camera control for text-to-video generation[J]. arXiv preprint arXiv:2404.02101, 2024.

[5] You M, Zhu Z, Liu H, et al. NVS-Solver: Video Diffusion Model as Zero-Shot Novel View Synthesizer[J]. arXiv preprint arXiv:2405.15364, 2024.

[6] Shi X, Huang Z, Wang F Y, et al. Motion-i2v: Consistent and controllable image-to-video generation with explicit motion modeling[C]//ACM SIGGRAPH 2024 Conference Papers. 2024: 1-11.

[7] Ku M, Wei C, Ren W, et al. Anyv2v: A plug-and-play framework for any video-to-video editing tasks[J]. arXiv preprint arXiv:2403.14468, 2024.

[8] Ouyang W, Dong Y, Yang L, et al. I2VEdit: First-Frame-Guided Video Editing via Image-to-Video Diffusion Models[J]. arXiv preprint arXiv:2405.16537, 2024.

[9] Ju X, Gao Y, Zhang Z, et al. Miradata: A large-scale video dataset with long durations and structured captions[J]. arXiv preprint arXiv:2407.06358, 2024.

[10] Peebles W, Xie S. Scalable diffusion models with transformers[C]//Proceedings of the IEEE/CVF International Conference on Computer Vision. 2023: 4195-4205.

[11] Perazzi F, Pont-Tuset J, McWilliams B, et al. A benchmark dataset and evaluation methodology for video object segmentation[C]//Proceedings of the IEEE conference on computer vision and pattern recognition. 2016: 724-732.

[12] Wu J, Li X, Zeng Y, et al. Motionbooth: Motion-aware customized text-to-video generation[J]. arXiv preprint arXiv:2406.17758, 2024.

[13] Hou C, Wei G, Zeng Y, et al. Training-free Camera Control for Video Generation[J]. arXiv preprint arXiv:2406.10126, 2024.

---

### Meta-Review · Area_Chair_1CVd · 2024-12-21

**Metareview:**

The paper presents Trajectory Attention, a method for precise camera motion control in video generation. It specifically introduces a trajectory-specific attention branch alongside temporal attention and uses pixel trajectories from optical flow to enhance motion precision and consistency. Experiments show significant improvements in video quality and adaptability to tasks like camera motion control and first-frame-guided editing, all without altering original model parameters.

Agreed by most reviewers, the strengths of Trajectory Attention include (1) its lightweight design, which integrates seamlessly with existing models without altering original parameters; (2) the novel incorporation of pixel trajectories through an auxiliary attention branch, enhancing motion precision and long-range consistency; and (3) its strong performance across diverse tasks, such as camera motion control and video editing, supported by thorough experiments and ablation studies.

The paper's weaknesses include (1) heavy reliance on dense optical flow, which may increase inference time and make real-time applications challenging; (2) limited evaluation on diverse datasets, as experiments are primarily conducted on MiraData, potentially restricting the generalizability; (3) insufficient exploration of performance with highly sparse, incomplete, or noisy trajectory inputs, which are common in real-world scenarios; and (4) limited comparisons with recent stoa methods, such as those applying epipolar geometry for better 3D consistency.

Despite the weaknesses, all reviewers expressed a positive overall rating, particularly after the rebuttal addressed key concerns. The ACs reached a consensus to accept the paper.

**Additional Comments On Reviewer Discussion:**

For the aforementioned weaknesses, the authors addressed them effectively: (1) they clarified the reasonable computational cost of dense optical flow and demonstrated support for sparse trajectories; (2) expanded evaluations to include adaptability to architectures like 3D DiTs, with qualitative results provided; (3) strengthened comparisons with recent methods, such as Collaborative Video Diffusion and Camco, while noting limitations due to unavailable code; (4) validated generalizability with new dataset analysis and included real-world examples for robustness; and (5) improved clarity in figures and trajectory extraction processes, ensuring a more comprehensive and balanced presentation. Limitations and future directions were also acknowledged.

---

### Decision · Program_Chairs · 2025-01-22

Accept (Poster)